

# Towards reducing the high cost of parameter sensitivity analysis in hydrologic modelling: a regional parameter sensitivity analysis approach

Samah Larabi[1], Juliane Mai[2], Markus Schnorbus[1], Bryan A. Tolson[2], Francis Zwiers[1]

[1]Pacific Climate Impacts Consortium, University of Victoria, Victoria, British Columbia, Canada
[2]University of Waterloo, Department of Civil and Environmental Engineering, Waterloo, Ontario, Canada

*Correspondence to*: Samah Larabi (slarabi@uvic.ca)

**Abstract.** Land surface models have many parameters that have a spatially variable impact on model outputs. In applying these models, sensitivity analysis (SA) is sometimes performed as an initial step to select calibration parameters. As these models are applied on large domains, performing sensitivity analysis across the domain is computationally prohibitive. Here, using a VIC deployment to a large domain as an example, we show that watershed classification based on climatic attributes and vegetation land cover helps to identify the spatial pattern of parameter sensitivity within the domain at a reduced cost. We evaluate the sensitivity of 44 VIC model parameters with regard to streamflow, evapotranspiration and snow water equivalent over 25 basins with a median size of 5078 km². Basins are clustered based on their climatic and land cover attributes. Performance of transferring parameter sensitivity between basins of the same cluster is evaluated by the *F1* score. Results show that two donor basins per cluster are sufficient to correctly identify sensitive parameters in a target basin, with *F1* scores ranging between 0.66 (evapotranspiration) to 1 (snow water equivalent). While climatic attributes are sufficient to identify sensitive parameters for streamflow and evapotranspiration, including vegetation class significantly improves skill in identifying sensitive parameters for snow water equivalent. This work reveals that there is opportunity to leverage climate and land cover attributes to greatly increase the efficiency of parameter sensitivity analysis and facilitate more rapid deployment of land surface models over large spatial domains.

## 1 Introduction

Land surface models (LSMs) are often used over large-scale domains (i.e., continental, or subcontinental river basins) to analyze hydrologic variables of interest. The main purpose of large-domain hydrologic modelling is to simulate, in a spatially consistent manner, the processes governing water fluxes across different geographic and hydroclimatic regions (Mizukami et al., 2017). The application of LSMs over large domains raises several challenges, including the availability of driving data and observations for calibration and the computational cost of calibration.



Parameter estimation when modelling the hydrology of large domains is particularly challenging due the number of parameters
that must be estimated, the resulting computational demand and the impact of spatial heterogeneity on parameter
transferability. Given the lack of guidance on parameter transferability over large domains, LSMs often rely on a priori
parameterizations based on expert opinion, case studies, field data, or hydrologic theory (Beck et al., 2016, Rakovec et al.,
2019). Specifically, LSM parametrization of vegetation and soil characteristics is generally based on other measured
characteristics or found in the literature from soil and vegetation classes (Nasonava et al., 2009). This approach relies on the
assumption that vegetation and soil type solely determine the ideal values of vegetation parameters and soil parameters
respectively, neither of which is supported by previous studies (e.g., Rosero et al., 2010; Cuntz et al., 2016; Bennett et al.,

37  2018).

LSM parameter estimation is a high dimensional problem (Göhler et al, 2013; Cuntz et al., 2016). The calibration parameter
space can, however, be reduced by a sensitivity analysis (SA) that serves to identify parameters that strongly influence the
model output variance. SA provides objective insights on calibration parameters by eliminating  parameters from the
calibration space that do not affect model output variance (hereafter called noninformative parameters) and reducing the
probability of over-parameterization (Van Griensven et al., 2006; Cuntz et al., 2015; Demirel et al., 2018). The computational
cost of SA depends on the number of model runs needed to simulate realistic model responses, which increases significantly
with the number of model parameters considered (Sarrazin et al., 2016; Devak and Dhanya, 2017). Therefore, SA of LSMs is
either overlooked and calibration parameters are selected based on the expert judgement and/or a previous SA, or when
performed, the list of model parameters analyzed is artificially shortened to exclude numerous model parameters whose values
are not known with certainty. Recent sensitivity analysis studies of LSMs, have however, revealed the impact of fixed-value
parameters (i.e., parameters assigned fixed values, often within the model code itself) on model output variance (e.g., Mendoza
et al., 2015; Cuntz et al., 2016; Houle et al., 2017), thus raising the need to explore and estimate these parameters to improve
the spatial accuracy of LSM outputs and the representation of hydrologic processes.
Sensitivity analysis studies show that parameter sensitivities vary geographically depending on the hydroclimatic conditions
(Demaria et al., 2007; Gou et al., 2020) and considered hydrologic processes (Bennett et al., 2018; Sepúlveda et al., 2021). As
land surface models are often applied on increasingly larger domains, performing sensitivity analysis across the entire domain
to identify the spatial pattern of sensitive parameters becomes increasingly computationally prohibitive, particularly when one
considers the large number of parameters involved. In addition, there is a lack of guidance in the literature on ways to
extrapolate parameter sensitivity from local to the larger scale with a reduced computational cost.
One approach for extrapolating parameter sensitivity is watershed classification, which aims at identifying watersheds that are
similar in some sense (i.e., according to certain attributes). Hydrological applications of watershed classification include
understanding general catchment hydrologic behavior (e.g., Sawicz et al., 2011), estimation of flow duration curves and
streamflow in ungauged sites (e.g., Boscarello et al., 2016; Kanishka and Eldho, 2020) and estimation of environmental model



parameters in scarce data regions (e.g., Jafarzadegan et al., 2020). In this paper, we investigate the utility of watershed
classification for reducing the cost of large-scale parameter sensitivity.
Our objective is to demonstrate the application of watershed classification as a means to regionalize parameter sensitivity. We
do this using an example deployment of the Variable Infiltration Capacity model (VIC, Liang et al., 1994, 1996). The VIC
model has been extensively used for regional hydrological modelling, but with typically only 4 to 11 parameters adjusted
during calibration (e.g., Wenger et al. 2010; Shreshta et al., 2012; Oubeidillah et al., 2013; Schnorbus et al., 2014; Islam et al.,
2017; Lohmann et al., 1998; Nijssen et al., 2001; Xie and Yuan, 2006; He and Pang, 2014; Melsen et al., 2016; Yanto et
al.,2017; Ismail et al., 2020; Gou et al., 2020; Waheed et al., 2020). Nevertheless, many additional VIC parameters that are
typically fixed also affect model output variance (e.g., Mendoza et al., 2015; Melsen et al., 2016; Houle et al., 2017; Bennett
et al., 2018). Hence, we examine the regionalization of parameter sensitivity for a much larger suite of 44 parameters that
includes 14 soil parameters, four climate parameters, six snow-related parameters, three glacier parameters and 17 vegetation
related parameters. In order to address a range of hydrologic processes, parameter sensitivity is assessed with regard to three
model outputs: streamflow, evapotranspiration and snow water equivalent.
This paper is organized as follows. Section 2 describes the study area, the VIC-GL model and its parametrization, the sequential
screening method and the watershed classification approach used. Section 3 presents the results of the sensitivity analysis for
streamflow, evapotranspiration, snow cover, and the results of transferring parameter sensitivity based on watershed
classification. Section 4 provides a discussion of the results followed by conclusions in Sect. 5, where we also discuss the
implications for cost effective sensitivity analysis when considering hydrologic models with large numbers of parameters that
are deployed across large domains.
**2 Methods**
Section 2.1 presents the study area and the dataset used to drive the VIC-GL model. Section 2.2 describes the version of VIC
used here, while Sec. 2.3 describes its parametrization and initialization. The parameter sampling strategy is also described in
Sect. 2.3. Section 2.4 presents the Efficient Elementary Effects (EEE; Morris, 1991) screening method used to identify VIC-
GL informative parameters. Section 2.5 presents the physical similarity approach used to transfer parameter importance to
other basins.
**2.1 Study area and dataset**
The study area extends over the Pacific Northwest region of North America from 40.75° N to 57.6° N and 109.96° W to 127.9°
W (see Fig.1). It encompasses three large watersheds, the Peace, Fraser and Columbia rivers, with a combined area of
1,150,624 km$^2$. This region spans many physiographic and climatic zones, resulting in substantial hydroclimatic spatial
variability. The domain was subdivided into several smaller basins (158 in total) according to location of hydrometric gauges.





We selected 25 of these basins representing glacierized conditions in the Coast Mountains and the Rocky Mountains, semi-
arid conditions in the interior of both the Fraser and Columbia and in eastern Peace, and the arid conditions of the southern
Columbia. The location of these basins is presented in Fig. 1 and their characteristics are summarised in Table 1 and 2. The
selected basins capture large spatial variability in precipitation, which is largely controlled by orography, such that average
annual precipitation over the 25 basins ranges from 448 mm/year to 1666 mm/year. The sampled basins also capture a strong
latitudinal gradient of air temperature, with average annual temperature ranging from -0.4 °C to 7.4 °C. The snow index, the
fraction of annual precipitation that falls as snow, ranges from 0.38 to 0.70 and the aridity index, the ratio of evapotranspiration
to precipitation (ET/P), ranges from 0.28 to 1.66. Average catchment elevation ranges from 683 m to 1990 m.

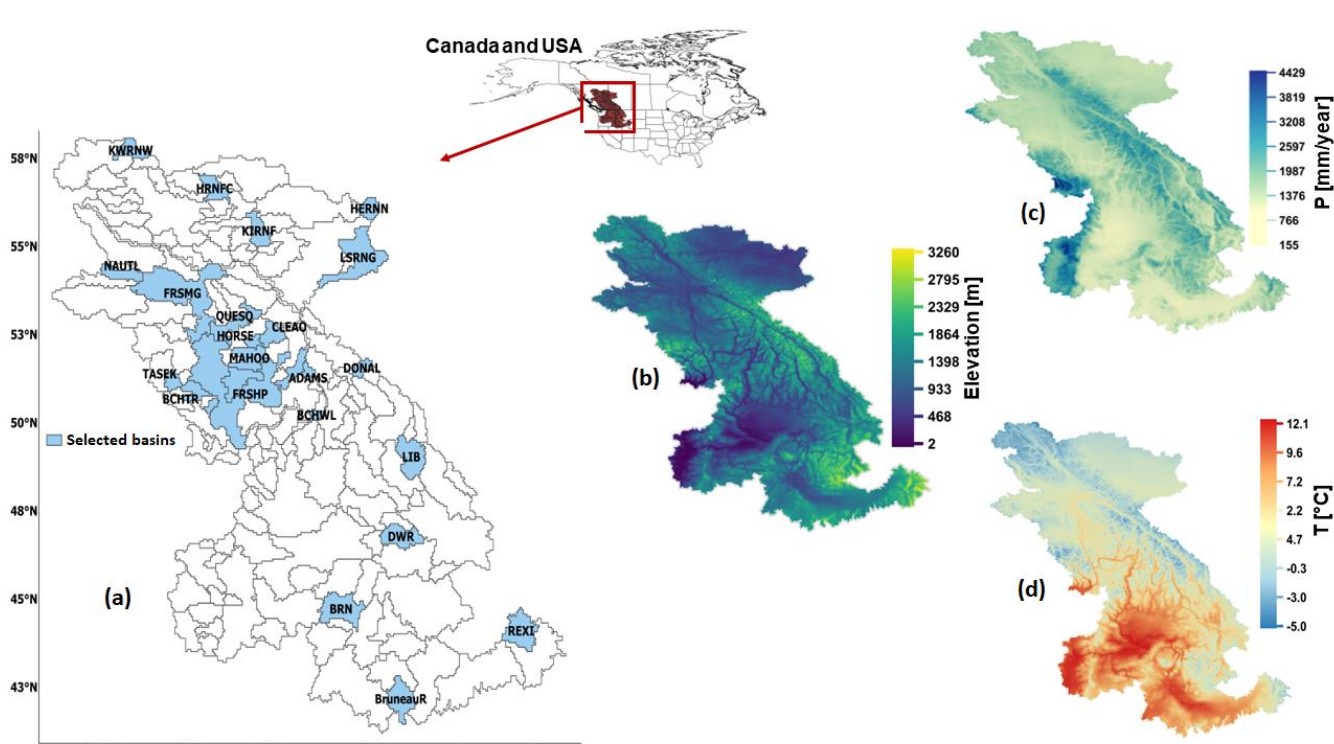

**Figure 1: Modelled domain with the location of the 25 selected sub-basins (a), the domain digital elevation map (b), mean annual**
**precipitation (c) and mean annual temperature (d), which were calculated from the PNWNAmet dataset.**






**Table 1: Physiographic attributes of 25 selected basins.**

| Basin ID | Basin name | Basin description | Area [km²] | Glacier area [km²] | Average elevation [m] | Relief [m] |
|---|---|---|---|---|---|---|
| 1 | ADAMS | Adams River near Squilax, BC | 3130 | 41 | 1266 | 1558 |
| 2 | BCHTR | Bridge River at Terzaghi Dam, BC | 2745 | 54 | 1748 | 1434 |
| 3 | BCHWL | Shuswap River at Wilsey Dam, BC | 1021 | 0 | 1339 | 1208 |
| 4 | BONAP | Bonaparte River below Cache Creek, BC | 5334 | 0 | 1216 | 1305 |
| 5 | BRN | Snake River at Brownlee Dam, Idaho/Oregon | 8877 | 0 | 1299 | 1692 |
| 6 | CAYOO | Cayoosh Creek near Lilooet, BC | 954 | 2 | 1770 | 1400 |
| 7 | CLEAO | Clearwater River at the outlet of Clearwater Lake, BC | 3031 | 224 | 1625 | 1540 |
| 8 | DONAL | Columbia River at Donald, BC | 1623 | 115 | 1767 | 1838 |
| 9 | DWR | North Fork Clearwater River at Dworshak Dam, ID | 6066 | 0 | 1307 | 1341 |
| 10 | FRSHP | Fraser River at Hope, BC | 31557 | 62 | 1198 | 2015 |
| 11 | FRSMG | Fraser near Marguerite, BC | 20810 | 0 | 867 | 968 |
| 12 | HERNN | Krawchuk Drainage near Mclennan, BC | 4018 | 0 | 683 | 160 |
| 13 | HORSE | Horsefly River above McKinley Creek, BC | 1242 | 0 | 1400 | 990 |
| 14 | KIRNF | Kiskatinaw River near Farmington, BC | 6196 | 0 | 910 | 555 |
| 15 | LIB | Kootenai River at Libby Dam, MT | 6977 | 0 | 1327 | 1240 |
| 16 | LSRNG | Little Smoky River near Guy, AB | 18975 | 0 | 868 | 946 |
| 17 | MAHOO | Maood River at outlet of Mahood Lake, BC | 5078 | 0 | 1194 | 1072 |
| 18 | NAUTL | Nautley river near Fort Fraser, BC | 3163 | 0 | 956 | 565 |
| 19 | QUESQ | Quesnel River near Quesnel, BC | 5551 | 78 | 1251 | 1442 |
| 20 | SEYMO | Seymour River near Seymor Arm, BC | 1024 | 41 | 1516 | 1422 |
| 21 | TASEK | Taseko River at outlet of Taseko Lake, BC | 1789 | 194 | 1990 | 1098 |
| 22 | REXI | Henrys Fork Rexburg, ID | 8034 | 0 | 1983 | 1590 |
| 23 | BurneauR | Bruneau River near Hot Spring, Idaho | 7074 | 0 | 1711 | 1852 |
| 24 | KWRNW | Kwadacha River Near Ware, BC | 5034 | 144 | 1538 | 1433 |
| 25 | HRNFC | Halfway River near Farrel Creek, BC | 5906 | 0 | 835 | 705 |



The climatic attributes presented in Table 2 are spatially averaged by sub-basin from the gridded PNWNAmet dataset (Werner
et al., 2019), which is used to drive the VIC model. This dataset provides gridded observations of daily precipitation (mm) and
minimum and maximum temperature (°C) for the Northwestern North America. The dataset is available at a daily timestep
and a spatial resolution of 1/16° for the period 1945 to 2012. Wind speed (m/s) from the 20CR reanalysis (Compo et al., 2011)
that has been spatially interpolated to 1/16° is also provided with the PNWNAmet dataset at a daily timescale. For further
details see Werner et al. (2019).
**Table 2: Climatic attributes of the 25 selected basins. Snow Index is the fraction of wet days when temperature is below 2°C, zero**
**means no snow and one means all precipitation is received as snow. Aridity index is the ratio between average annual**
**evapotranspiration and precipitation (ET/P).**

| Basin name | Average annual precipitation [mm] | Average annual temperature [°C] | Snow index | Aridity index |
|---|---|---|---|---|
| ADAMS | 1196 | 3.39 | 0.47 | 0.40 |
| BCHTR | 1123 | 1.42 | 0.62 | 0.37 |
| BCHWL | 991 | 3.64 | 0.51 | 0.48 |
| BONAP | 475 | 3.88 | 0.43 | 1.04 |
| BRN | 557 | 7.42 | 0.40 | 1.01 |
| CAYOO | 995 | 1.93 | 0.60 | 0.43 |
| CLEAO | 1492 | 1.00 | 0.57 | 0.28 |
| DONAL | 1194 | 0.23 | 0.61 | 0.34 |
| DWR | 1271 | 5.88 | 0.48 | 0.41 |
| FRSHP | 951 | 3.96 | 0.44 | 0.51 |
| FRSMG | 634 | 2.94 | 0.44 | 0.76 |
| HERNN | 448 | 1.23 | 0.47 | 1.14 |
| HORSE | 1119 | 2.17 | 0.51 | 0.40 |
| KIRNF | 575 | 2.19 | 0.45 | 0.87 |
| LIB | 856 | 3.93 | 0.48 | 0.56 |
| LSRNG | 570 | 2.62 | 0.41 | 0.90 |
| MAHOO | 675 | 3.34 | 0.45 | 0.72 |
| NAUTL | 583 | 2.64 | 0.45 | 0.82 |
| QUESQ | 939 | 2.86 | 0.46 | 0.50 |
| SEYMO | 1666 | 2.63 | 0.70 | 0.28 |
| TASEK | 1310 | -0.37 | 0.70 | 0.29 |
| REXI | 729 | 3.54 | 0.54 | 0.65 |
| BruneauR | 337 | 7.43 | 0.38 | 1.66 |
| KWRNW | 845 | -1.57 | 0.62 | 0.47 |
| HRNFC | 514 | 1.61 | 0.48 | 0.96 |





## 2.2 VIC-GL model

VIC is a physically based macroscale model that simulates both water and energy balances by grid cells (Liang et al., 1994, 1996; Cherkauer and Lettenmaier, 1999). The VIC model has been widely applied to analyze the impact of climate change on the hydrology and water resources of the study region (e.g., Hamlet and Lettenmaier, 1999; Payne et al., 2004; Shrestha et al., 2012; Schnorbus et al., 2014; Islam et al., 2017) and to study the effect of land cover change on streamflow (e.g., Matheussen et al., 2000). VIC-GL, an upgraded version developed at the Pacific Climate Impacts Consortium (PCIC) that is used here, includes additional functionality to simulate glacier mass balance (Schnorbus, 2018). VIC-GL was branched from VIC version 4.2, and although the model physics are in many ways similar, it uses a different model abstraction from its predecessor. Although the computational domain of VIC-GL is still described using a two-dimensional grid (using a spatial resolution of 1/16° in the current application), sub-grid variability in land cover and topography uses hydrologic response units (HRUs) as opposed to the original vegetation tiles. Specifically, an HRU is assigned for each land cover class within an elevation band, with the elevation of each HRU being the median of the associated elevation band. In this manner, the type and extent of land cover is allowed to vary with elevation within grid boxes. The vertical water and energy balance is solved separately in each HRU and then averaged to the grid-cell scale. The current application of VIC-GL uses fixed 200-m elevation bands and three soil layers. The baseline model processes are described in detail by Liang et al. (1994, 1996), Cherkauer et al. (2013) and Bohn et al. (2016).

Updates to address glacier mass balance modelling are described in detail by Schnorbus (2018), but pertinent VIC-GL parameter changes are summarised here. Glacier surface mass and energy balance modelling introduces three additional parameters *GLAC_ALB*, *GLAC_ROUGH* and *GLAC_REDF*. *GLAC_ALB* specifies the albedo of glacier ice, which controls the amount of incoming solar radiation absorbed by the ice surface. The value of *GLAC_ALB*, once set, is constant in time. The parameter *GLAC_ROUGH* specifies the roughness length of the glacier surface, which affects the wind speed profile and the transfer of energy to the glacier surface due to the turbulent fluxes. The scaling factor for snow redistribution (*GLAC_REDF*) controls the redistribution of precipitation between non-glacier HRUs and acts as a proxy for mechanical snow redistribution that typically occurs via wind and gravity in mountainous alpine environments (e.g. Kuhn 2003). VIC-GL also uses the rain-snow partitioning algorithm of Kienzle (2008) rather than the original algorithm in the VIC model distribution. This is a curvilinear model that uses two parameters, the threshold mean daily temperature (*TEMP_TH_1*), where 50% of precipitation falls as snow, and the temperature range centered on *TEMP_TH_1* within which both solid and liquid precipitation occurs (*TEMP_TH_2*). VIC-GL has also been updated to make certain parameters more accessible for model calibration and to allow for a more spatially explicit description of some hydro-climatic processes. These parameters include five that determine soil albedo decay according to the USACE algorithm (USACE 1956) and the climatic parameters *T_LAPSE* and *PGRAD*. The latter specify vertical temperature and the precipitation gradients that are used to adjust temperature and precipitation, respectively, for each HRU within a grid cell.



## 2.3 Model parameterization and sampling

We consider 44 VIC-GL parameters (Table 3) composed of 5 baseflow parameters, 1 runoff parameter, 9 drainage parameters, 4 climate parameters, 6 snow-related parameters, 3 glacier parameters and 17 vegetation related parameters. The set of analyzed parameters includes the commonly calibrated parameters, parameters that have been addressed in previous studies (e.g., Demaria et al., 2007; Houle et al., 2017; Bennett et al., 2018), and some that are typically set to fixed values (Gao et al., 2009).

Table 3: The 44 VIC-GL parameters selected for the sensitivity analysis. Type is the parameter sampling strategy, which is to either replace the parameter default value (i.e., Absolute), apply a multiplicative factor or apply an additive change to the baseline values. The additive change is applied so that trunk ratio remains between 0.1 and 0.8.

| Parameter | Description | Unit | Range | Default | Type |
|---|---|---|---|---|---|
| **Baseflow parameters** | | | | | |
| ds | Fraction of Dsmax where nonlinear baseflow begins | _ | [0.001, 0.6] | 0.1 | Absolute |
| dsmax | Maximum velocity of baseflow | mm/day | [1, 200] | 40 | Absolute |
| ws | Fraction of maximum soil moisture where nonlinear baseflow occurs | _ | [0.4, 1] | 0.9 | Absolute |
| c | Exponent used in baseflow curve | _ | [1, 10] | 2 | Absolute |
| depth3 | Thickness of soil layer 3 | m | [0.5, 10] | 2 | Absolute |
| **Runoff parameters** | | | | | |
| INFIL | Variable infiltration curve parameter | _ | [0.0001, 0.8] | 0.2 | Absolute |
| **Drainage parameters** | | | | | |
| watn | Exponent in Campbell's equation for hydraulic conductivity in all layers | _ | [8, 11] | 9.5 | Absolute |
| ks | Saturated hydrologic conductivity in all layers | mm/day | [300, 3000] | 1081 | Absolute |
| depth1 | Thickness of soil layer 1 | m | [0.001, 0.5] | 0.1 | Absolute |
| depth2 | Thickness of soil layer 2 | m | [0.05, 1] | 0.2 | Absolute |
| bd | Soil bulk density (applied to all layers) | kg/m^3 | [800, 1600] | 1400 | Absolute |
| sdens | Soil particle density (applied to all layers) | kg/m^3 | [2000, 2700] | 2500 | Absolute |
| wcr | Critical Point (applied to all layers) | _ | [0.35, 0.55] | 0.40 | Absolute |
| wpwp | Wilting point (applied to all layers) | _ | [0.20, 0.50] | 0.35 | Absolute |
| resid_moist | Residual moisture (applied to all layers) | _ | [0.0, 0.125] | 0.08 | Absolute |
| **Climate parameters** | | | | | |
| PGRAD | Precipitation gradient | 1/m | [0.0001, 0.001] | 0.0005 | Absolute |
| T_LAPSE | Temperature lapse rate | °C/m | [0, 9.5] | 6.5 | Absolute |



| | | | | | |
|---|---|---|---|---|---|
| TEMP_TH_1 | Rain/snow temperature threshold parameter 1 | °C | [-2.0, 5.0] | 2 | Absolute |
| TEMP_TH_2 | Rain/snow temperature threshold parameter 2 | °C | [8.0, 15.0] | 12 | Absolute |
| **Snow parameters** | | | | | |
| SNOWROUGH | Surface roughness of snowpack | m | [0.0001, 0.1] | 0.01 | Absolute |
| NEW_SNOW_ALB | Albedo of new snow | _ | [0.8, 0.9] | 0.85 | Absolute |
| SNOW_ALB_ACCUM_A | Albedo decay coefficient during accumulation period | _ | [0.3, 0.99] | 0.94 | Absolute |
| SNOW_ALB_ACCUM_B | Albedo decay exponent during accumulation period | _ | [0, 0.99] | 0.58 | Absolute |
| SNOW_ALB_THAW_A | Albedo decay coefficient during thaw period | _ | [0.1, 0.99] | 0.82 | Absolute |
| SNOW_ALB_THAW_B | Albedo decay exponent during thaw period | _ | [0, 0.99] | 0.46 | Absolute |
| **Glacier parameters** | | | | | |
| GLAC_ALB | Albedo of glacier surface | _ | [0.2, 0.6] | 0.4 | Absolute |
| GLAC_ROUGH | Surface roughness of glacier | m | [0.0001, 0.01] | 0.001 | Absolute |
| GLAC_REDF | Scaling factor for snow redistribution with values in range 0 (no redistribution) to 1 (redistribution equal to area ratio) | _ | [0, 1] | 0 | Absolute |
| **Vegetation parameters** | | | | | |
| root_depth | Thickness of root zone layer 3 | m | [0.5, 2] | 1 | Multiplicative factor |
| root_fract1 | Fraction of roots in soil layer 1 | _ | [0, 1] | 0.7 | Absolute |
| root_fract2 | Fraction of roots in soil layer 2 | _ | [0, 1] | 0.2 | Absolute |
| lai_djf | Leaf Area Index (winter) | m2/m2 | [0.5, 2] | 1 | Multiplicative factor |
| lai_mam | Leaf Area Index (spring) | m2/m2 | [0.5, 2] | 1 | Multiplicative factor |
| lai_jja | Leaf Area Index (summer) | m2/m2 | [0.5, 2] | 1 | Multiplicative factor |
| lai_son | Leaf Area Index (fall) | m2/m2 | [0.5, 2] | 1 | Multiplicative factor |
| alb_dja | albedo(winter) | _ | [0.5, 2] | 1 | Multiplicative factor |
| alb_mam | albedo(spring) | _ | [0.5, 2] | 1 | Multiplicative factor |
| alb_jja | albedo(summer) | _ | [0.5, 2] | 1 | Multiplicative factor |
| alb_son | albedo(fall) | _ | [0.5, 2] | 1 | Multiplicative factor |
| Rarc | Architectural resistance | s/m | [0.5, 2] | 1 | Multiplicative factor |
| Rmin | Minimum stomatal resistance | s/m | [0.5, 2] | 1 | Multiplicative factor |





| RGL | Minimum incoming shortwave radiation at which there will be transpiration | W/m^2 | [0.5, 2] | 1 | Multiplicative factor |
|---|---|---|---|---|---|
| SolAtn | Solar attenuation factor | _ | [0.5, 2] | 1 | Multiplicative factor |
| WndAtn | Wind speed attenuation through the overstory | _ | [0.5, 2] | 1 | Multiplicative factor |
| Trunk_ratio* | Ratio of total tree height that is trunk | _ | [-0.2, 0.2] | 0 | Additive change |


The commonly calibrated parameters are limited to four baseflow parameters, the runoff parameter, and five drainage
parameters. The common baseflow parameters are maximum velocity of baseflow (*dsmax*), fraction of *dsmax* where nonlinear
baseflow begins *(ds)*, fraction of maximum soil moisture where non-linear baseflow occurs (*ws*) and thickness of deepest soil
layer *(depth3)*. These parameters describe the non-linear relationship between baseflow rate and soil moisture in the deepest
soil layer (with thickness described by *depth3*). The runoff parameter, or variable infiltration curve parameter *(INFIL)*,
describes the extent of soil saturation within grid cell (i.e., amount of direct runoff) as function of soil moisture in the surface
soil layers (i.e., the variable infiltration curve, Liang et al., 1994) which have thicknesses given by *depth1* and *depth2*. The
common drainage parameters are the two parameters controlling soil storage capacity (*depth1* and *depth2*), the exponent in
Campbell's equation for hydraulic conductivity (*watn*) and the saturated hydrologic conductivity (*ks*).
The additional drainage parameters considered are the soil bulk density (*bd*), soil particle density (*sdens*), fractional soil
moisture content at the critical point (wcr), fractional soil moisture content at the wilting point (*wpwp*) and the residual moisture
(*resid_moist*). The *wpwp* parameter dictates baseflow estimation with the Arno model formulation (Francini and Pacciani,
1991) used in VIC (Gao et al., 2009). We also consider the four climate parameters which are temperature lapse rate
(*T_LAPSE*), precipitation gradient, and the rain/snow temperature threshold parameter 1 and 2 (*TEMP_TH_1 and*
*TEMP_TH_2)*. The examined parameters also include the three glacier mass balance parameters (*GLAC_ALB, GLAC_ROUGH*
*and GLAC_REDF*). The snow related parameters examined are surface roughness (*SNOWROUGH*), albedo of new snow
(*NEW_SNOW_ALB*) and albedo decay parameters during the accumulation period (*SNOW_ALB_ACCUM_A,*
*SNOW_ALB_ACCUM_B*) and during the thaw period (*SNOW_ALB_THAW_A, SNOW_ALB_THAW_B*).
The parameters describing snow and glacier properties along with soil and climate parameters are assigned by grid cell. These
parameters were initialized with default values and then sampled within prescribed ranges (see Table 3).The same value is
assigned to all grid cells within a catchment. The sampling of the soil parameters critical point (*wcr*), wilting point (*wpwp*) and
residual moisture (*resid_moist*) is constrained so that conditions required by VIC (Gao et al., 2009) are not violated. Thus,
sampling is performed so that $wcr \leq (1 - bd/sdens)$, $wpwp \leq wcr$, and $resid\_moist \leq wpwp * (1 - bd/sdens)$.





The vegetation parameters consist of the thickness of root zone of the third soil layer (*root_depth*), and the root fractions in all
three soil layers. We only sample root fractions in soil layer one and two (*root_fract1*, *root_fract2)* such that the total root
fraction in the three soil layers adds to 1. That is, the *root fraction* in soil layer three is updated as 1 - (*root_fract1 + root_fract2*).
The vegetation parameters that are considered also include the seasonal leaf area index (*lai*) and seasonal albedo (*albedo*), the
architectural resistance (*Rarc*), minimum stomatal resistance (*Rmin*), minimum incoming shortwave radiation at which there
will be transpiration (*RGL*), solar attenuation factor (*SolAtn*), wind speed attenuation through the overstory (*WndAtn*) and
fraction of the total tree height that is occupied by tree trunks (*Trunk_ratio*). The *lai* parameter governs the amount of water
intercepted by the canopy, which controls canopy evaporation. Leaf area index, along with stomatal resistance (*Rmin*), also
influences the estimation of vegetation transpiration, and the root fraction dictates the amount of transpiration from each soil
layer (Gao et al., 2009). The parameter *Rarc* affects the vertical wind profile.
The vegetation parameters are assigned by land cover class. Sampling of these parameters is conducted by adjusting baseline
values obtained for each land cover class. The land cover classes were based on the North America Land Cover dataset,
edition2 (Natural Resources Canada/The Canada Centre for Mapping and Earth Observation (NRCan/CCMEO) et al. 2013)
produced as part of the North America Land Change Monitoring System (NALCMS). In total, 22 land cover classes were
identified. For most of these parameters, sampling is conducted by applying a multiplication factor, sampled in the range 0.5
to 2.0, to the baseline values. The same sampled parameter is applied to all vegetation classes. To reduce the number of
vegetation parameters, a multiplier factor is applied on a seasonal basis for the monthly parameters *LAI* and *albedo*, following
a similar approach of Bennett et al., (2018). For example, *lai_djf* is the multiplier factor applied to leaf area index values during
winter months (i.e., December, January, and February). The *trunk ratio* is sampled around the defined value by applying an
additive change in the range -0.2 to 0.2 so that *trunk ratio* values remain between 0.1 and 0.8. The monthly roughness and
displacement height parameters were not sampled. They are specified as a function of vegetation height (which is constant
within classes, but variable between classes) and leaf area index as described by Choudhury and Monteith (1988).

**2.4 Sensitivity analysis**

We applied the Efficient Elementary Effects (EEE) screening method introduced by Cuntz et al. (2015) as a frugal
implementation of the Morris method (Morris, 1991). It was developed to identify the model parameters that are most
informative regarding a certain model output. The strength of the method lies in it requiring only a small set of model
evaluations to separate informative vs. noninformative parameters. On average, EEE requires $10N$ model runs with $N$ being
the number of model parameters. EEE does not require algorithmic tuning and converges by itself. The method has been tested
for a large range of sensitivity benchmarking functions and a hydrologic model at several locations by Cuntz et al. (2015). The
method has further been applied to obtain the informative parameters in complex hydrologic (Cuntz et al., 2016) and land-
surface models (Demirel et al., 2018).





The EEE approach samples model parameters in trajectories as initially described by Morris (1991) and improved by
Campolongo et al. (2007). A "trajectory" is defined as a sequence of ($N$+1) parameter sets where the first parameter set is
sampled randomly while all subsequent sets $i$ ($i > 1$) differ from the prior set ($i$-1) in exactly one parameter value. Such
trajectories allow an efficient sampling of the whole parameter space while considering parameter interactions to a certain
extent. In the approach of Cuntz et al. (2015), only a small number of such trajectories ($M_1$; here $M_1$=5) are sampled in a first
EEE iteration to lower the computational burden. The resulting ($M_1$ x ($N$ + 1)) model outputs are derived, and the elementary
effects (EEs) are computed for each parameter following Morris (1991). The EEs are used to identify the most informative
parameters by deriving a threshold that splits the parameters into a set of $N_{ninf}$ noninformative parameters and a set of $N_{inf}$=$N$-
$N_{ninf}$ informative parameters. The threshold $T$ is derived automatically within the EEE method and is based on the EEs of the
model outputs provided in the first iteration. The threshold is derived based on fitting a logistic function to the sorted EEs
derived and defining the threshold as the point of largest curvature of the fitted logistic function. Defining the threshold that is
used to separate informative and non-informative parameters in this approach has been demonstrated using a wide range of
test functions and real-world examples, and the reader is referred to Cuntz et al. (2015) for further details. In the next EEE
iteration, a new $N$-dimensional parameter set is randomly sampled but this time only the $N_{ninf}$ noninformative parameters are
perturbed while the $N_{inf}$ informative parameters are kept at their initially sampled values. Hence, this trajectory contains only
$N_{ninf}$+1 parameter sets. $M_2$ of such trajectories are sampled in this step (here $M_2$=1). The derivation of model outputs and the
calculation of EEs is repeated. If the EE of any noninformative parameter exceeds the previously derived threshold $T$, the
previously noninformative parameter will be added to the set of informative parameters. This EEE iteration (sampling a new
trajectory and then adding parameters with an EE above $T$ to the set of informative parameters) is repeated until no further
parameter is reclassified as informative. The final EEE iteration is to sample $M_3$ trajectories (here $M_3$=5) to confirm that the
set of $N_{ninf}$ noninformative parameters is stable, and no further parameter is found to be informative. The EEE method parameter
values ($M_1$, $M_2$, and $M_3$) utilized here are the default settings tested and recommended by Cuntz et al. (2015). The
implementation, documentation, and examples for EEE are open source (Mai and Cuntz, 2020).
**2.5 Transferability of parameter sensitivity**
We applied the EEE method to each of the 25 basins and the three model outputs (streamflow, evaporation, snow water
equivalent) independently, leading to 75 sets of noninformative/informative parameters. The initial set of $N$ randomly sampled
model parameter values was the same for all 75 experiments. An average of 430 model runs were required for each of the 75
EEE experiments to identify which of the 44 VIC-GL parameters analyzed in this study were informative.
Informative and noninformative parameters were compared over the 25 basins to identify parameters that are informative
across all basins (termed invariant-informative parameters), 2) parameters that are non-informative across all basins (invariant-
noninformative, and 3) parameters that are informative in some basins but not others (variant-informative).





We evaluated the potential of using watershed classification as a tool to transfer parameter SA information. Climatic conditions
exert a major control on runoff generation (Yadav et al., 2007; Sawics et al., 2011) and have been found to have a higher
impact on parameter sensitivity than vegetation and soil conditions (Rosero et al., 2010). However, vegetation and soil
conditions can affect other hydrologic quantities. For example, Bennett et al. (2018) found that canopy spacing plays an
important role in snow water equivalent simulation by VIC. Here, we used aridity index, snow index and the percentage of
glacier area, and the percentage of area covered by each of several vegetation classes to classify the 25 basins. Although 22
vegetation classes are defined for VIC-GL, we only considered the four vegetation classes listed in Table 4 that are dominant
in the study area. To evaluate the impact of vegetation on informative parameter identification, watershed classification was
first performed using the climatic attributes only, and then by combining climatic and vegetation class cover attributes.

**Table 4: Statistics of the percentage of VIC land cover classes (%) identified using NALCMS and considered in this study over the**
**25 selected basins.**

| Class ID | Description | Min | Max | Mean |
|---|---|---|---|---|
| 2 | Temperate or sub-polar needleleaf forest - high-elevation | 0.1 | 46 | 18 |
| 4 | Temperate or sub-polar needleleaf forest - coastal/humid/dense | 0 | 29 | 9 |
| 9 | Mixed Forest | 0 | 34 | 4 |
| 11 | Temperate or sub-polar shrubland | 0.4 | 91 | 19 |


To classify the 25 basins into homogenous groups, the agglomerative hierarchical algorithm was used with the Euclidean
distance and Ward's criterion (Roux, 2018). Agglomerative hierarchical clustering consists of a series of successive fusion of
watersheds into groups according to their similarity. It starts by considering each element x (i.e., watershed) as a cluster {x}
then continue by creating new cluster by merging the two closest clusters. The dendogram, a tree diagram, illustrates the
merging process of the agglomerative hierarchical clustering. The Ward method used here aggregates clusters so that within-
group inertia (i.e. multidimensional variance) is minimal.
To test our hypothesis that parameter sensitivity can be generalized using watershed classification we conducted the following
evaluation. Each sub-basin was set as the target basin. For each target basin, informative parameters are transferred using a
number of donor basins of the same cluster. Using multiple donor basins has been shown to provide better results than a single
donor basin (e.g. Oudin et al., 2008; Bao et al., 2012). Let A be a target basin of cluster $C_i$. We assume that informative
parameters of basin A are the intersection of informative parameters of x donor basins from cluster $C_i$. For each target basin
A, informative parameters are transferred using all possible combinations of x donor basins of cluster $C_i$ not including A. This
test aims at evaluating whether x donor basins could be used to generalize informative parameters for each cluster.





The performance of watershed classification to identify informative and noninformative parameters in a basin is evaluated
using the *F1* score. This score is often used to measure the performance of a binary classification (Chicco and Jurman, 2020).
The *F1* score is a weighted average of precision and recall. Assuming two classes, positive (informative) and negative
(noninformative), the *F1* score measures the ability to correctly and incorrectly predict the two classes. Considering counts of
TP true positive (i.e., informative predicted as informative), FP false positive (informative predicted as noninformative), and
FN false negative (noninformative predicted as informative), we can obtain measures of precision, recall and the *F1* score as
follows:
$Precision = \frac{TP}{TP+FP}$,  (1)
$Recall = \frac{TP}{TP+FN}$ ,  (2)
$F1\ score = 2 * \frac{Precision \times Recall}{Precision + Recall}$  (3)

The *F1* score takes values between 0 and 1, where 0 means that all positive (here informative parameters) are predicted as
negative (i.e., as noninformative) and 1 means perfect classification with *FN=FP=0*.
For a given number of donor basins x, the *F1* score is reported for each target basin A as the average *F1* score calculated
between sensitive parameters of A and identified sensitive parameters from all possible combinations of the x donor basins.
This is done for each classification method, climate-based and climate-land cover-based clustering, to evaluate performance
in identifying sensitive parameters by watershed groupings provided by each clustering analysis. Then, we use the Wilcoxon
signed rank test to compare the *F1* scores for the 25 basins obtained using the two clustering methods so that we can determine
whether incorporating land cover in watershed classification improves the ability to predict informative parameters. The
Wilcoxon signed rank test tests the null hypothesis that the *F1* score resulting from both clustering analyses are from the same
distribution i.e., have similar ability to identify informative parameters.
**3 Results**
The sensitivity analysis using the EEE method was performed with respect to three model outputs independently: streamflow,
evapotranspiration, and snow water equivalent. Figure 2 presents the number of occurrences of informative parameters over
the 25 selected sub-basins for the three outputs. From this figure, we can identify the three parameter categories, invariant-
informative, invariant noninformative and variant-informative for each hydrologic process. Table 5 summarizes the three
parameter categories per model output. Amongst the 44 VIC-GL parameters only 9 parameters are invariant-informative for
streamflow, 13 are invariant-informative for evapotranspiration and 4 are invariant-informative for snow water equivalent. A
large percentage of parameters are variant-informative for these fluxes with 29 parameters for streamflow, 25 parameters for
evapotranspiration and 14 parameters for snow water equivalent. We first examine the sensitive parameters and their spatial





variability per model output in Sect. 3.1 to 3.3. We further analyze the performance of the physical similarity approach for
transferring sensitivity analysis information and the attributes that are informative for each model output (Sect. 3.4).

**Figure 2: Number of occurrences of informative parameters for streamflow (a), evapotranspiration (b) and snow water equivalent**
**(c) over the 25 studied sub-basins. Parameters are considered invariant-informative if the count of basins in which they are**
**informative**





**Table 5: VIC-GL parameter importance regarding streamflow, evapotranspiration (ET) and snow water equivalent (SWE).**

| Process | Invariant-informative parameters | Invariant-noninformative parameters | Variant-informative parameters |
|---|---|---|---|
| **Streamflow** | ds, dsmax, ws, depth3, INFIL, depth1, bd, sdens, resid_moist | PGRAD, GLAC_ROUGH, alb_mam, alb_jja, alb_son, RGL | c, T_LAPSE, watn, ks, depth2, wcr, wpwp, SNOW_ROUGH, NEW_SNOW_ALB, SNOW_ALB_ACCUM_A, SNOW_ALB_ACCUM_B, SNOW_ALB_THAW_A, SNOW_ALB_THAW_B, TEMP_TH_1, TEMP_TH_2, GLAC_ALB, GLAC_REDF, root_depth, root_fract1, root_fract2, lai_djf, lai_mam, lai_jja, lai_son, alb_dja, Rarc, Rmin, Sol_Atn, Trunk_ratio |
| **ET** | depth1, depth2, bd, wcr, wpwp, resid_moist, TEMP_TH1, TEMP_TH2, root_fract1, root_fract2, lai_mam, lai_jja, Rmin | SNOW_ALB_THAW_B, GLAC_ALB, GLAC_ROUGH, GLAC_REDF, alb_dja, alb_son | ds, dsmax, ws, c, depth3, INFIL, PGRAD, T_LAPSE, watn, ks, sdens, SNOW_ROUGH, NEW_SNOW_ALB, SNOW_ALB_ACCUM_A, SNOW_ALB_ACCUM_B, SNOW_ALB_THAW_A, root_depth, lai_djf, lai_son, alb_mam, alb_jja, Rarc, RGL, Sol_Atn, Trunk_ratio |
| **SWE** | SNOW_ROUGH, NEW_SNOW_ALB, SNOW_ALB_THAW_A, TEMP_TH1 | ds, dsmax, ws, c, depth3, INFIL, watn, ks, depth2, bd, sdens, wcr, wpwp, resid_moist, GLAC_ALB, GLAC_ROUGH, GLAC_REDF, root_depth, root_fract1, root_fract2, alb_dja, alb_jja, alb_son, Rarc, Rmin, RGL, | PGRAD, T_LAPSE, depth1, SNOW_ALB_ACCUM_A, SNOW_ALB_ACCUM_B, SNOW_ALB_THAW_B, TEMP_TH_2, lai_djf, lai_mam, lai_jja, lai_son, alb_mam, Sol_Atn, Trunk_ratio |

## 3.1 Informative parameters for streamflow

The soil parameters *ds*, *dsmax*, *ws*, *depth3*, *depth1* are consistently identified as sensitive to streamflow (e.g., Demaria et al., 2007; Bennett et al., 2018; Gou et al., 2020) and this reflects the empirical nature of the runoff and baseflow processes that are fundamental in the VIC family of models. In addition to these parameters, the soil parameters soil bulk density (*bd*), soil particle density (*sdens*) and the residual moisture (*resid_moist*) are also identified as invariant-informative to streamflow in the study area.

Figure 3 presents the sensitivity of the 29 variant-sensitive parameters with respect to streamflow (Table 5). These parameters include the remaining soil parameters, climate, snow, and most of the vegetation parameters. The climate parameters *TEMP_TH_1 and TEMP_TH_2* (i.e., the rain/snow temperature threshold parameter 1 and 2) have different sensitivity patterns. The parameter *TEMP_TH_1* is found to be informative across all basins except in the arid basin BruneauR, which has the lowest snow index (0.38). The parameter *TEMP_TH_2* is informative only in sub-basins located in the interiors of the Fraser and Peace. *T_LAPSE* is informative in the snow-dominated basins of the Fraser and the Columbia. The snow-related





parameters show different spatial sensitivity. For instance, *SNOW_ROUGH* is sensitive over all basins except for some snow-
dominated basins of the Fraser and Columbia. The *NEW_SNOW_ALB* and *SNOW_ALB_THAW_A*, which control snow melt,
are sensitive across all basins except the semi-arid basins of the Peace (north-east of the study region). Snowmelt in the study
area contributes significantly to runoff, which explains the sensitivity of these parameters for streamflow. These results are
consistent with the results found by Houle et al. (2017) who evaluated sensitivity of these parameters to snow water equivalent
using the Sobol' method (Sobol', 1990).

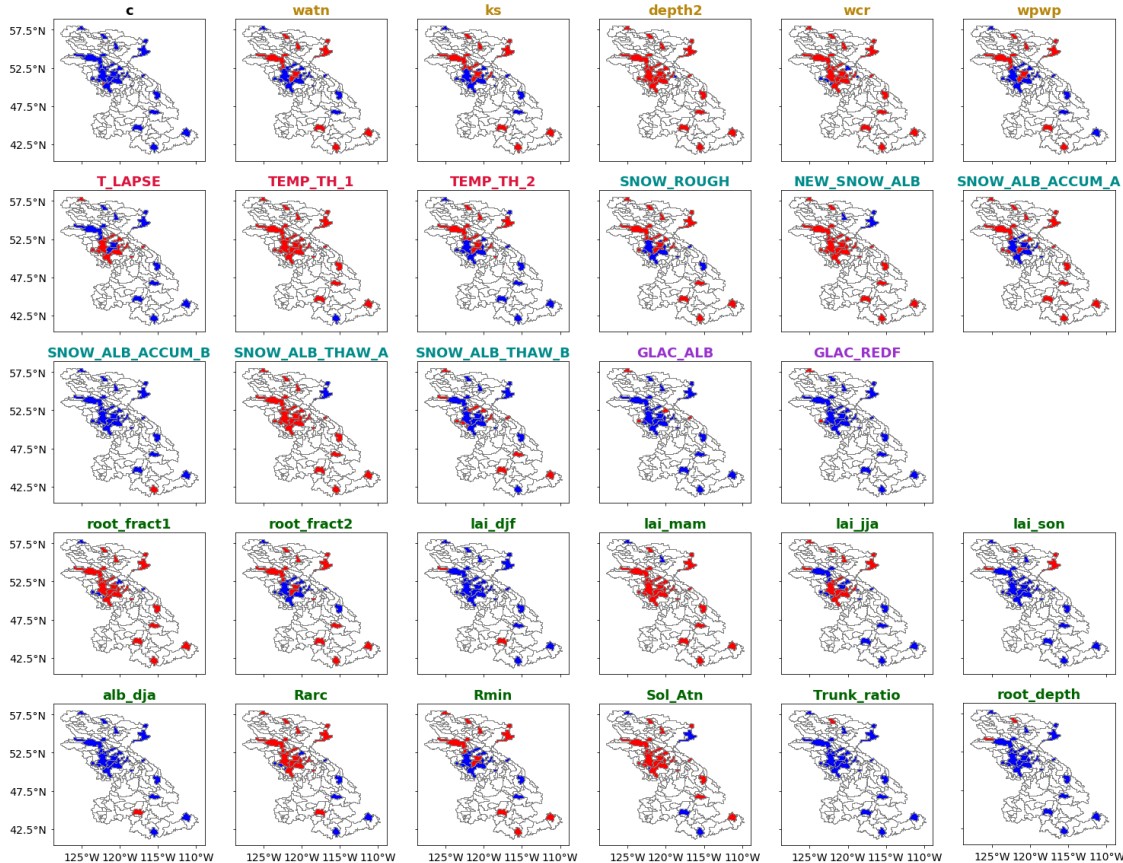


**Figure 3: The spatial sensitivity of the 29 streamflow variant-informative parameters with red being informative and blue non-informative over the 25 selected basins. The nine invariant informative and six invariant non-informative parameters are not included.**

In the semi-arid and arid basins, the exponent in Campbell's equation for hydraulic conductivity (*watn*), the saturated
hydrologic conductivity (*ks*), and fractional soil moisture content at the wilting point (*wpwp*) are informative for streamflow.
The *wpwp* parameter dictates baseflow estimation with the Arno model formulation (Francini and Pacciani, 1991) used in VIC
(Gao et al., 2009). Given the limited precipitation in these basins, baseflow may be a significant streamflow source that explains
the importance of this parameter in these basins. The root depth of the third layer (*root_depth*) is sensitive in the northern semi-





arid basins (NAUTL, HRNFC). The root fraction of the first layer (*root_fract1*) is sensitive in Columbia basins and the non-
glacierized basins of the Fraser and Peace. The root fraction in the second layer (*root_fract2*) is sensitive only in the semi-arid
and arid basins. The sensitivity of the LAI parameters is seasonal with springtime LAI being sensitive in almost all basins.
For the glacierized headwater catchments the albedo of the glacier surface (*GLAC_ALB)* is informative for streamflow. The
importance of this parameter increases with the basin glacier area and this parameter is influential in the four basins CLEAO,
KWRNW, DONAL, and TASEK with the largest glacier area (between 115 km$^2$ and 194 km$^2$, between 7 % and 11 % of
watershed area). The remaining glacierized basins have much smaller glacier areas (less than 1.5 % of the watershed area).
The *GLAC_REDF* parameter is informative for streamflow as well in the western-glaciated basins TASEK and KWRNW,
where average annual temperature is negative. Glaciers behave as natural water reservoirs that provide streamflow through ice
melt and temporary meltwater storage within the glacier during late summer (Marshall et al., 2011). For instance, in the upper
Columbia, glaciers contribute up to 25 % and 35 % of streamflow in August and September respectively and up to 6 % to the
annual streamflow (Jost et al 2012, Jiskoot and Muller, 2012).

**3.2 Informative parameters for evapotranspiration**

There are 13 invariant-informative parameters that affect evapotranspiration in the study region (see Fig. 2 and Table 5). These
include parameters that control soil drainage (*wcr*, *wpwp*, *resid_moist*), and soil storage capacity (*bd*, *depth1* and *depth2*). The
invariant-informative parameters also include the climate parameters *TEMP_TH_1*, *TEMP_TH_2* and vegetation parameters
seasonal leaf area index (*lai_mam*, *lai_jja*), minimum stomatal resistance (*Rmin*), and root fraction (*root_fract1*, *root_fract2*).
The VIC-GL model computes evapotranspiration as the sum of four types of evaporation; evaporation from the canopy layer,
transpiration from all three soil layers, soil evaporation from the top soil layer, and evaporation/sublimation from the snow or
glacier surface (Liang et al., 1994). The soil parameters affect the bare soil evaporation that occurs at the top thin layer. The
leaf area index parameters govern the amount of water intercepted by the canopy, which controls canopy evaporation. Leaf
area index and stomatal resistance (*Rmin*) influence the estimation of vegetation transpiration and the root fraction dictates the
amount of transpiration from each soil layer (Gao et al., 2009). These parameters are defined for each land cover type in the
vegetation library. They are typically fixed based on observed values, which ignores the large estimation and scaling
uncertainties around their values (Mendoza et al., 2015). In this paper, the sampling of *LAI* and *Rmin* values is based on a
perturbation of observed values (see Table 3; Type "Multiplicative factor"). The sensitivity of evapotranspiration to this
perturbation illustrates the need to obtain accurate values for these parameters or consider their uncertainty in the model
calibration process. The rain/snow temperature thresholds (*TEMP_TH_1*, *TEMP_TH_2*) are likely to impact the throughfall
(water that penetrates a plant canopy) and rainfall/snow interception (rain captured, stored, and evaporated from the vegetation
surface) (Levia et al., 2019).

**Figure 4: The spatial sensitivity of the 25 evapotranspiration variant-informative parameters with red being informative and blue non-informative over the 25 selected basins. The 13 invariant informative and 6 invariant non-informative parameters are not included. For the number of occurrences of informative parameters see Figure 2.**

Table 5 lists the six invariant-noninformative parameters for evapotranspiration which are the glacier parameters, autumn and winter vegetation albedo, and the albedo decay exponent during the thaw period *SNOW_ALB_THAW_B*. Figure 4 presents the spatial sensitivity of the 25 variant-informative parameters with respect to evapotranspiration. Some parameters show a clear spatial pattern of sensitivity that is related to basin physical characteristics. For instance, *T_LAPSE* is sensitive in snow-dominated basins, whereas *INFIL* and *sdens* are sensitive in semi-arid and arid basins. The baseflow parameters (*ds*, *dsmax*) are informative in most basins while the parameter *ws* is only informative in humid sub-basins. The surface roughness of the





snowpack (*SNOW_ROUGH*), the architectural resistance of vegetation (*Rarc*), which affects the vertical wind profile, and
autumn leaf area index (*lai_son*) are also influential to evapotranspiration in most basins.

**3.3 Informative parameters for snow water equivalent**

Amongst the six snow-parameters, only three (*SNOW_ROUGH*, *NEW_SNOW_ALB*, *SNOW_ALB_THAW_A)* are invariant-
informative for snow water equivalent. The climate parameter *TEMP_TH_1* is also invariant-informative for snow water
equivalent. The parameter *TEMP_TH_2* is informative in the majority of the basins except in the semi-arid basins of the Peace.
The sensitivity of the remaining three snow parameters (*SNOW_ALB_ACCUM_A*, *SNOW_ALB_ACCUM_B*, *and*
*SNOW_ALB_THAW_B*) and the two climate parameters (*PGRAD*, *T_LAPSE*) varies within the study region. Figure 5 presents
the sensitivity of the 14 variant-informative parameters for snow water equivalent. The *T_LAPSE* and *PGRAD* are sensitive in
the high-altitude basins. The parameter *SNOW_ALB_ACCUM_B* is informative in the basins of the Columbia and Peace, and
in the semi-arid basins of the Fraser. The sensitivities of seasonal leaf area index (*lai_djf*, *lai_mam*, *lai_jja*, and *lai_son*), ratio
of total tree height that is trunk (*Trunk_ratio*), and the solar attenuation factor (*Sol_Atn*) show a clear spatial pattern. These
parameters are informative in basins where forest is the dominant land cover (i.e., Fraser and Peace). The springtime vegetation
albedo (*alb_mam*) is sensitive over the snow-dominated basins. The sensitivity of snow water equivalent for vegetation
parameters can be explained by the impact of forest cover on snow accumulation and ablation processes, mainly by snowfall
interception and modification of incoming radiation and wind speed below the forest canopy (Andreadis et al., 2009). These
finding are consistent with those of Houle et al., (2017) and Bennett et al., (2018).





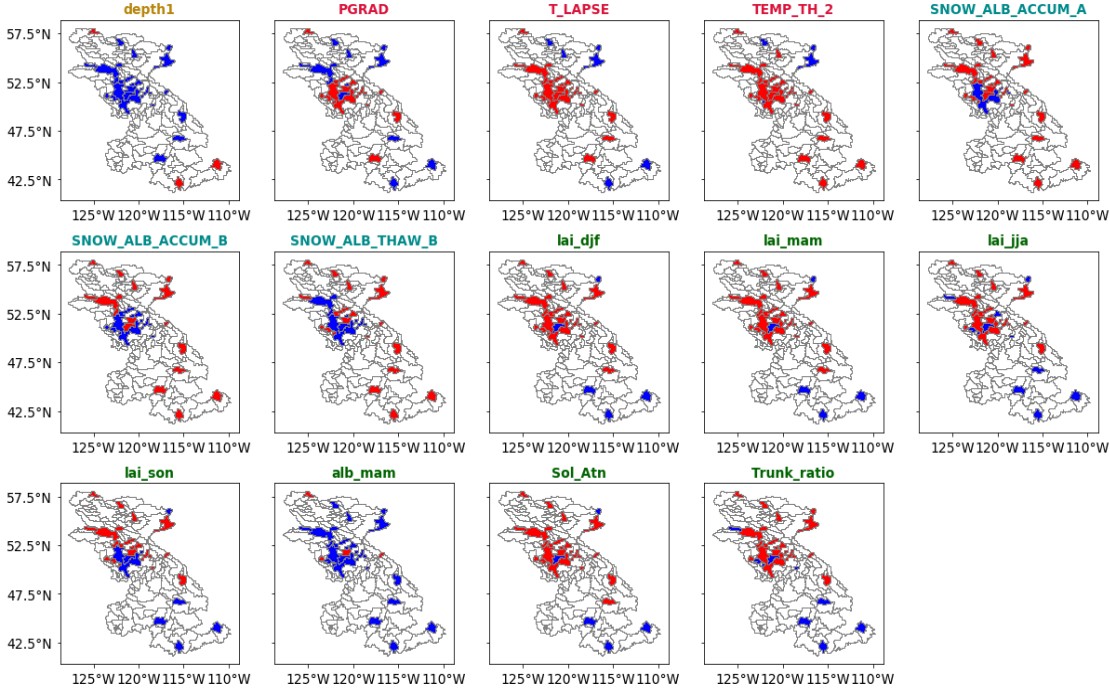

**Figure 5: The spatial sensitivity of the14 snow water equivalent variant-informative parameters with red being informative and blue non-informative over the 25 selected basins. The 4 invariant informative and 26 invariant non-informative parameters are not included. For the number of occurrences of informative parameters see Figure 2.**

**3.4 Watershed classification**

Figure 6 presents the dendogram, a diagram tree of clusters resulting from the agglomerative hierarchical clustering using climate indices and the combination of climate indices and vegetation class cover. Clustering based on climate indices yields four clusters whereas clustering based on climate indices and vegetation cover results in five clusters.





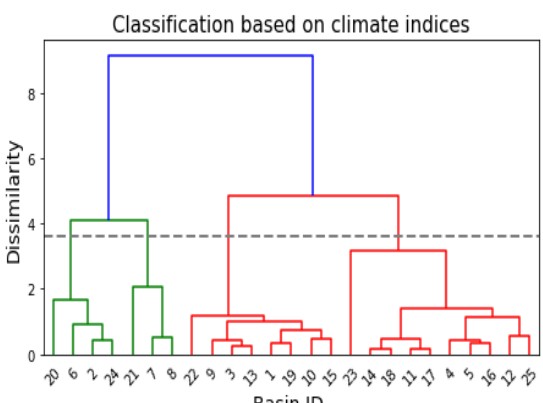 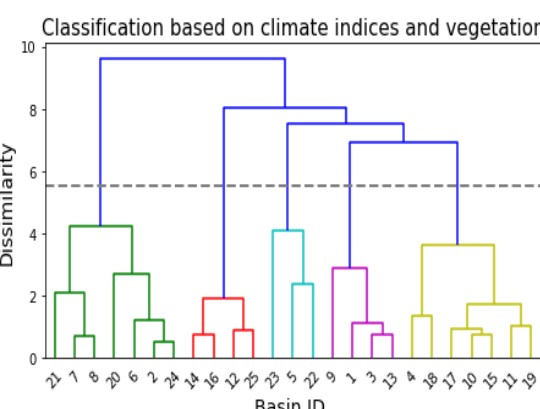

**Figure 6: Watershed classification dendogram using climate indices and the combination of climate and vegetation indices. The height of each node represents the distance between its branches and the dashed line represents the cutoff threshold to distinguish the 4 clusters in the case of climate-based classification and 5 clusters in the case of climate-land cover-based classification. The threshold is chosen as a trade-off between cluster dissimilarity and within cluster variance.**

Figure 7 shows the results of the hierarchical clustering analyses and Fig. 8 and 9 present the attribute statistics for each cluster. The clusters produced using climatic attributes can be described as follows. Cluster #1 consists of dry basins located in the southern Columbia, eastern Peace, and central Fraser basins. Cluster #2 contains glacierized watersheds along the Coast Mountains and the Rocky Mountains. Cluster #3 contains semi-arid basins in the interior Fraser and eastern Columbia, and cluster #4 contains snow-dominated basins with very low glacier area (less than 4 % of watershed area) compared to cluster #2. Clusters obtained using both climatic and vegetation attributes correspond to clusters based on climate that were merged or divided based on vegetation class cover dominance. Cluster #1 contains all glaciered watersheds and corresponds to clusters #2 and #4 obtained with climatic based clustering. Cluster #2 consist of dry basins dominated by land cover 11 (temperate or sub-polar shrubland) that are located in the southern Columbia basin. Cluster #3 consist of dry basins dominated by land cover 9 (i.e., mixed forest) located in the eastern Peace River basin. Cluster #4 represents arid basins in the interior Fraser and upper Columbia dominated by land cover 2 (i.e., temperate or sub-polar needleleaf forest - high-elevation) and cluster #5 consists of wet basins dominated with land cover 4 (i.e., temperate or sub-polar needleleaf forest - coastal/humid/dense).





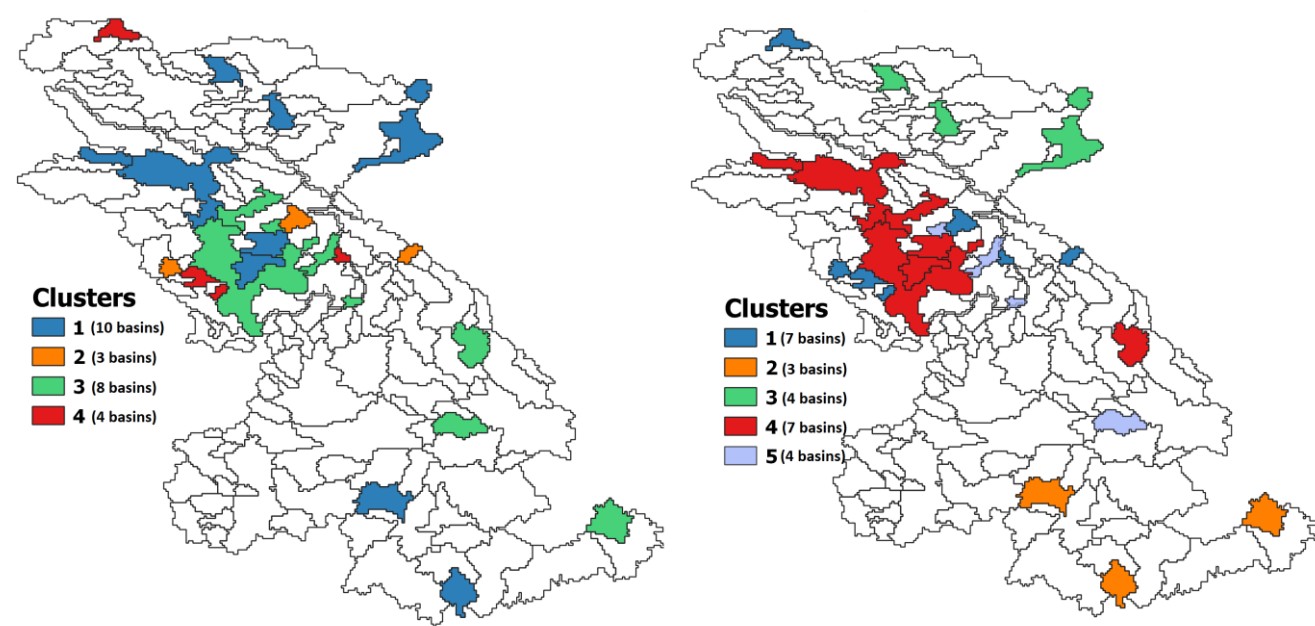


**Figure 7: Map of clusters obtained using only climatic attributes (left), and using both vegetation- and climatic attributes (right).**




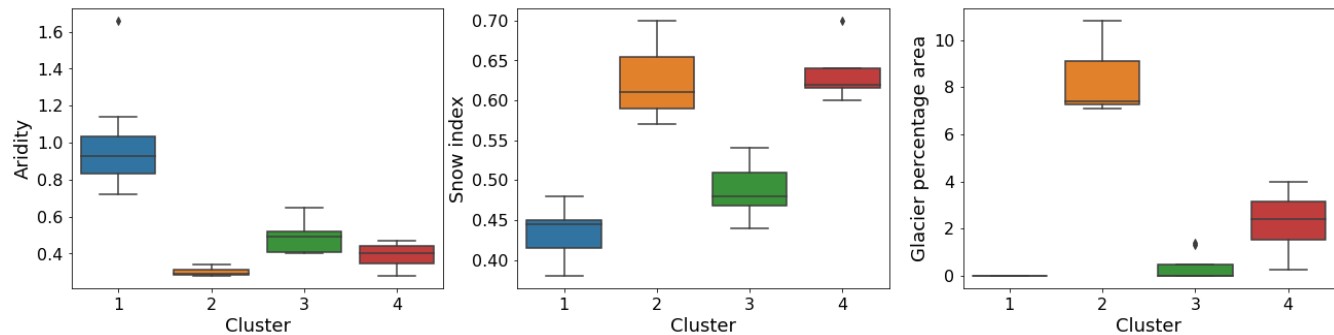


**Figure 8: Box-plots of the climate attributes for each cluster produced by climate based classification.**



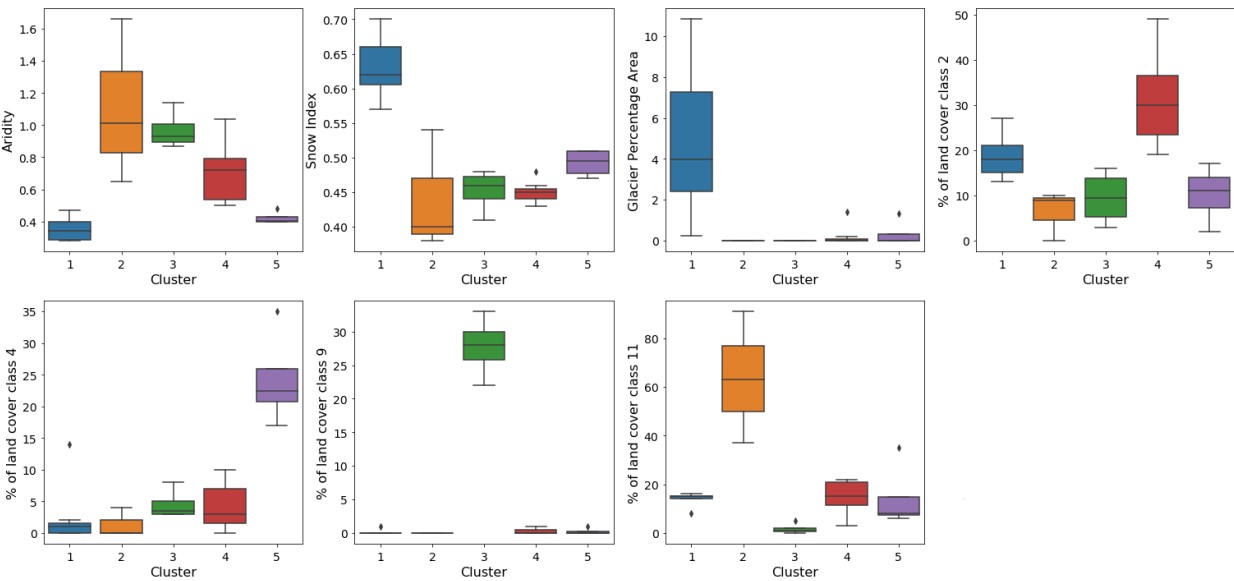

**Figure 9: Box-plots of attributes of each cluster produced by climate- and vegetation-based classification.**

## 3.5 Watershed classification as a way to transfer parameter sensitivity

The distribution of *F1* scores obtained by transferring informative parameters for streamflow, evaporation and snow water equivalent using both clustering analyses and a range of donor basins is presented in Fig. 10. The *F1* scores calculated for transferring streamflow informative parameters based on climatic attributes range between 0.66 (using 9 donor basins) and 0.98 (using between three to seven donor basins), whereas this score ranges between 0.65 (using six donor basins) and 0.96 (using six donor basins) when using both climate and vegetation attributes. For evapotranspiration the *F1* scores obtained by climatic based clustering range between 0.63 (using six donor basins) and 0.96 (using three to six donor basins). The scores range between 0.7 (using two donor basins) and 0.95 (using a single donor basin) when using both climatic and land cover attributes for clustering analysis. The *F1* scores for snow water equivalent range between 0.83 (using four to nine donor basins) and 1 (using one to two donor basins) when transferring informative parameters based on climatic attributes and the combination of climatic attributes and vegetation.





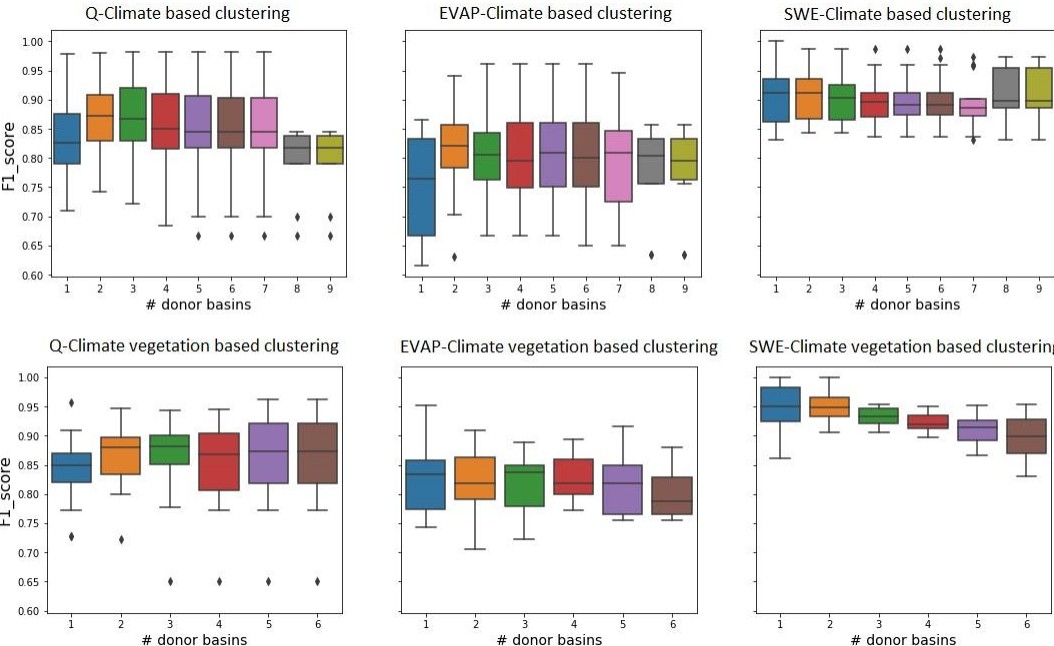

**Figure 10:** *F1* **score distribution obtained by transferring informative parameters over the 25 basins.**

Transferring informative parameters based on more than a single donor basin improves the *F1* score except when transferring evapotranspiration informative parameters using climatic and vegetation clustering analysis. Overall, the results shows that two donor basins would be sufficient to generalize informative parameters to each cluster. Therefore, for each model output we compare the *F1* distributions using two donor basins based on both clustering analysis with the Wilcoxon test. The p-value of the test applied to *F1* score distributions obtained by transferring streamflow informative parameters is 0.49 and by transferring evapotranspiration informative parameters is 0.48. Hence, the *F1* score distributions using climatic clustering analysis and climatic-land cover analysis are not significantly different. Therefore, using only climatic attributes would be sufficient to transfer informative parameters to streamflow and evapotranspiration. These findings are consistent with other VIC studies (Demaria et al., 2007) and for other hydrologic models (e.g., Rosero et al., 2010) showing that parameter sensitivity for streamflow can be transferred based predominantly on climate similarity.

The Wilcoxon test statistic applied to the *F1* distribution resulting from transferring snow water equivalent informative parameters is 31 with a p-value of 0.0006. This suggests that there is a significant improvement when using both climatic and land cover attributes to transfer snow water equivalent parameter sensitivity. The importance of land cover and vegetation properties as a control on snow accumulation and ablation is consistent with previous studies (e.g., Bennett et al., 2018).





## 4 Discussion

In this work, we have examined the sensitivity of an extensive list of VIC parameters to streamflow, evapotranspiration, and snow water equivalent over 25 basins spanning a range of hydroclimatic conditions. We found that informative parameters vary spatially with climate and land cover depending on the model output considered. The findings are in line with previous VIC sensitivity analysis studies (e.g., Demaria et al., 2007; Bennett et al., 2018; Gou et al., 2020, Sepúlveda, 2021). In addition, the two climate parameters temperature lapse rate (*T_LAPSE*) and the precipitation gradient (*PGRAD*) omitted in previous studies have been found to be informative to headwater glacierized watersheds and snow dominated non-glacierized watersheds. The *T_LAPSE* parameter is typically fixed when developing gridded meteorological data. For instance, Bohn et al., (2016) used a gridded temperature corrected with a lapse rate of 6.5 °K/km to force VIC over southwestern US and northwestern Mexico. However, several studies have indicated that the often-used constant lapse rates 6-6.5 °C/km are not representative of the surface conditions over different mountainous regions and may differ for each slope within the same mountain (Blandford et al., 2008; Minder et al., 2010, Córdova et al., 2016).

In this study, we showed that watershed classification helps identify spatial patterns of informative parameters at a reduced cost. Hence, it reduces the cost of performing sensitivity analysis at the same scale of large-scale land surface models. In our case, watershed classification based on climatic attributes (snow and aridity index) and percentage of glacier area was sufficient to transfer parameter sensitivity between basins of similar attributes. However, incorporating vegetation class cover significantly improved the identification of sensitive parameters for snow water equivalent. The results show that two donor basins per cluster are sufficient to identify sensitive parameters. These results imply that the cost of running sensitivity analysis over a large domain encompassing N clusters of basins would be reduced to the cost of running 2N sensitivity analyses. The information gained can then be extrapolated to large domain based on sub-watershed membership to the N clusters. Thus, candidate parameters for model calibration can be identified at a substantially reduced computational cost as compared to running a large-domain sensitivity analysis. For example, climatic based classification of the 158 basins that covers the entire domain results in four watershed clusters (see Fig. 11) as follows. Cluster #1 consist of glaciered basins along the Coast Mountains and Rocky Mountains. Cluster #2 groups dry basins located in interior and southern Columbia, eastern Peace, and upper Fraser basins. Cluster #3 contains snow-dominated basins in north Peace River basin and eastern Columbia River basin whereas Cluster #4 contains rainfall dominated basins in western Columbia River basin. These clusters are consistent with the clusters obtained by classifying the 25 basins except for cluster #4 because the sample of the studied basins does not include any rainfall-dominated basins. Hence, the cost of performing a sensitivity analysis across the 158 basins is reduced to the cost of evaluating parameter sensitivity over eight basins (i.e., two basins for each basin cluster).



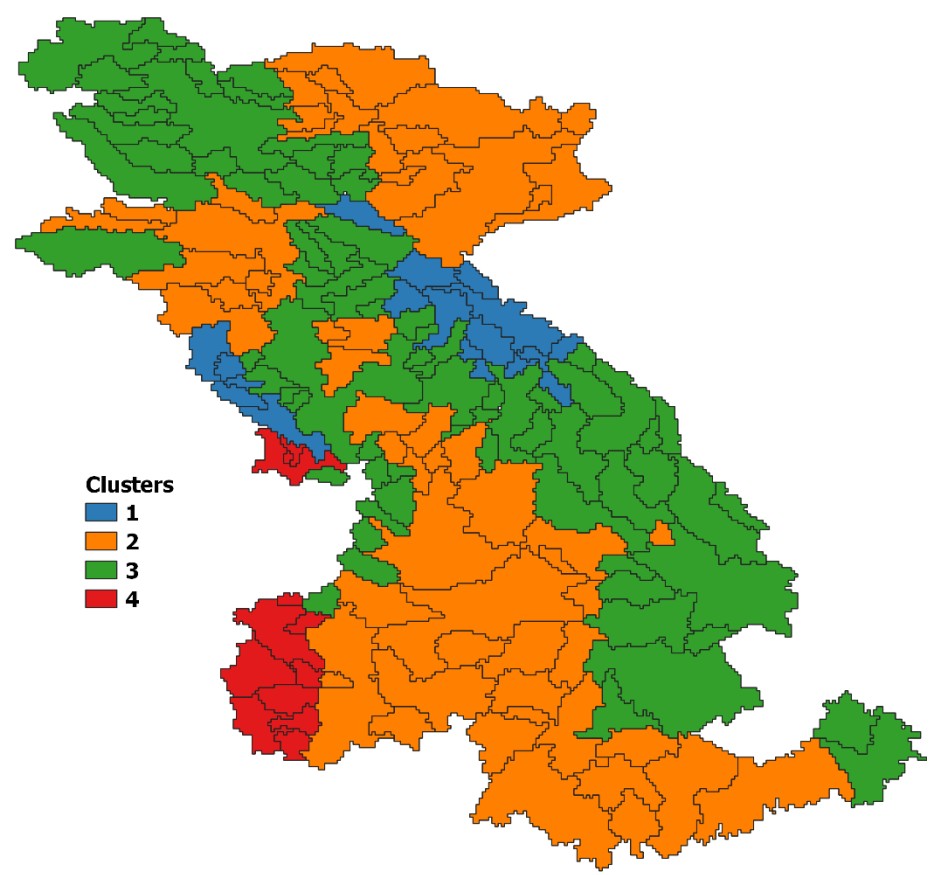


**Figure 11: Climatic based classification of the 158 sub-basins of the Peace River, Fraser River, and Columbia River basins.**


It has been argued in the literature that calibration based solely on streamflow is not sufficient to ensure model accuracy and
fidelity (Rakovec et al., 2016). To improve model realism, recent calibration strategies follow a process-based approach. This
approach relies either on adjusting model parameters against hydrological signatures extracted from streamflow timeseries that
link to the underlying model processes (Yilmaz et al., 2008; Euser et al., 2013, Shafii and Tolson; 2015; Rakovec et al., 2016),
against measurements of different model outputs such as evapotranspiration, snow cover, and baseflow (e.g., Isenstein et al.,
2015, Ismail et al., 2020), or by hydrograph decomposition (e.g., He et al., 2015, Shafii et al., 2017; Larabi et al., 2018).
However, we recognize that the effort to constrain multiple hydrologic processes will require a substantial increase in the size
of the parameter domain during model calibration. For instance, our sensitivity analysis results from Table 5 and Fig. 12
suggest that calibrating VIC-GL in a multi-objective/multi-variable framework would require a high number of parameters in
the calibration process (30 to 38 parameters depending on the sub-basin if one is to consider all informative parameters for
each output considered here).  Across the 25 sub-basins, an average of 77 % of parameters (34 of 44 parameters analyzed) are





informative to at least one of simulated streamflow, evapotranspiration, or snow water equivalent (see Fig. 12). This contrasts
with previous studies that typically calibrate fewer than 12 VIC parameters (e.g., Troy et al., 2008; Isenstein et al., 2015;
Mizukami et al., 2017; Rakovec et al., 2019; Ismail et al., 2020). Options to tackle this more complex calibration problem are
not evaluated here but could include suitable one-step multi-objective optimization algorithms such as PADDS (Asadzadeh et
al. (2014)), or a stepwise multi-objective calibration approach where each set of informative parameters for a specific flux are
adjusted separately (Larabi et al., 2018).

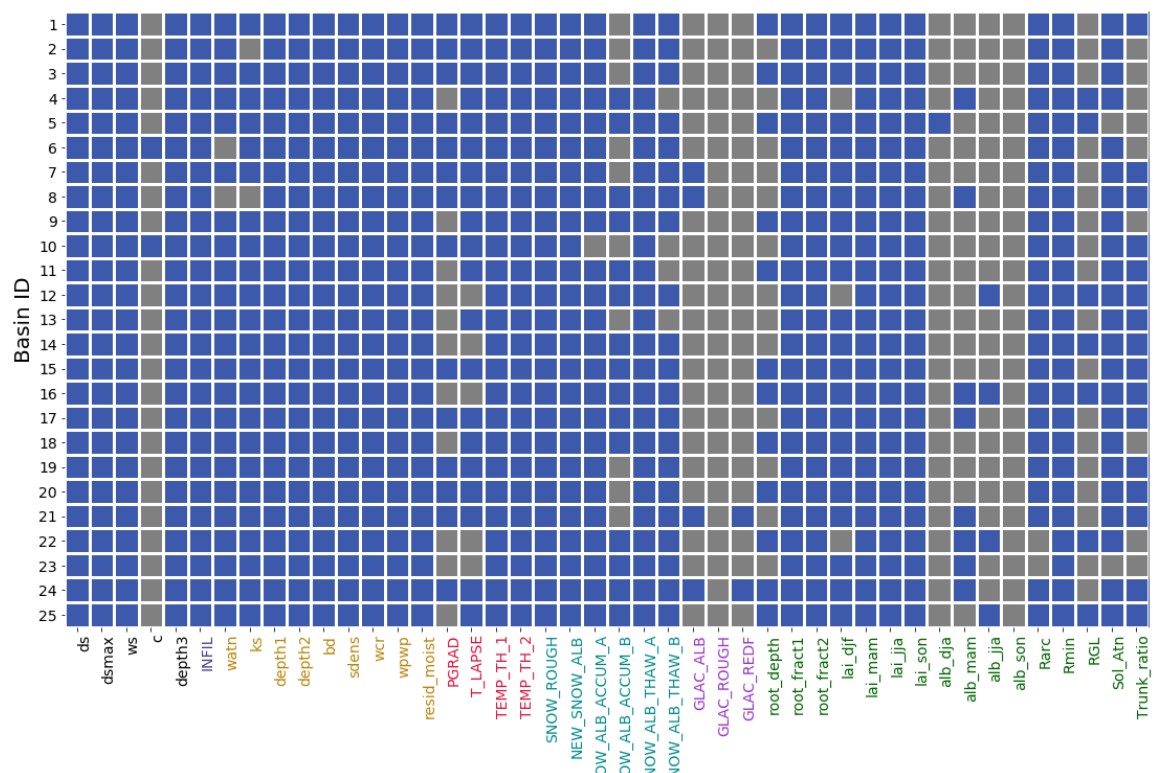


**Figure 12: Informative parameters (blue) for at least one of simulated streamflow, evapotranspiration, and snow water equivalent.**
**Basin ID description is provided in Table 1.**







In previous VIC applications, the same parameters are adjusted over large domains to fit the model to streamflow (e.g., Nijssen
et al., 2001; Obeidillah et al., 2014; Xue et al., 2015, Mizukami et al., 2017) and against other model output (Isenstein et al.,
2015; Ismail et al., 2020) ignoring both the spatial variability of parameter sensitivity and dependence of parameter sensitivity
to the hydrological processes. To account for this spatial variability, a multi-site cascading approach (Xue et al., 2015) where
calibration parameter selection varies depending on the site can be used. Overall, there remains a need to study how information
regarding the spatial variability and process dependence of parameter sensitivity is best integrated into a multi-variable
parameter estimation framework.
In this study, the low-cost EEE sequential screening method (Cuntz et al., 2105) was used to identify informative parameters.
However, this method does not quantitatively rank the importance of these informative parameters. In situations where it is
desired to reduce the number of calibration parameters below the counts identified by EEE analyses, a quantitative approach
such as variance-based methods (e.g., Sobol', 1990; Saltelli, 2002) or qualitative approach that provides parameter groupings
based on their sensitivity could be considered (Sheikholeslami et al., 2019; Mai et al., 2020, 2022). However, future work is
required to determine the conditions under which a reduction in the number of calibrated parameters (i.e., by not calibrating
some parameters that are informative) could potentially yield better calibration results, particularly in a multi-objective context.
**5 Conclusions**
Land surface models tend to have large numbers of parameters, many of which cannot be measured directly. Sensitivity
analysis is therefore often employed to identify parameters with significant impact on model output variance. Performing
sensitivity analysis for large-scale land surface models is, however, computationally demanding. In this study, we consider
whether computational cost can be reduced by using watershed classification to transfer information about which parameters
sensitively affect streamflow, evapotranspiration and snow water equivalent between basins that have similar climatic and
vegetation land cover attributes.
The study was performed using a large domain implementation of a hydrologic model as an example. Specifically, we used an
updated version of the VIC model (Schnorbus, 2018) that has been coupled to a regional glacier model and implemented across
a very large domain in the Pacific Northwest region of North America. A wide range of VIC model parameters was evaluated
that include five baseflow parameters, one runoff parameter, nine drainage parameters, four climate parameters, six snow-
related parameters, three glacier parameters, and 17 vegetation related parameters. The sensitivity analysis was performed over
25 basins spanning a range of hydroclimatic conditions to understand the spatial variability of parameter sensitivities with
regard to streamflow, evapotranspiration and snow water equivalent. Parameter sensitivities for each model output were found
to vary in a predictable way with basin climate and land cover characteristics.





Watershed classification was employed to classify the 25 basins into homogenous groups based on climatic attributes (aridity
and snow index) and percentage of glacier area and vegetation land cover. This classification was used to transfer sensitive
parameters to each basin based on its group membership. This approach was shown to be able to efficiently identify sensitive
parameters with a median *F1* score of 0.87 for streamflow, 0.83 for evapotranspiration and 0.95 for snow water equivalent.
These findings suggest that parameter sensitivity can be performed by classifying watersheds into broad groups and then
analyzing sensitivity for only a subset of the basins in each group. In our large domain example, we found that it would likely
be sufficient to perform sensitivity analysis in 4 % (or fewer) of the basins contained within the domain. This would
substantially reduce the cost of the sensitivity analyses that are used to determine the model calibration strategy, or for a given
computing budget, would enable the consideration of a broader range of parameters than could be considered if sensitivity
analysis were to be performed across the entire domain.
The parameter classification based on parameter sensitivities informs which parameters should be adjusted (invariant-
informative and variant-informative) depending on the calibration variables that are considered and the local climatic
conditions. We found that for a multi-variable calibration approach targeting streamflow, evapotranspiration and snow water
equivalent, an average of 77 % of VIC parameters (i.e., 34 of 44 parameters analyzed) were identified as calibration candidates.
These parameters include not only those that control runoff and baseflow generation, but also parameters that control snow
processes and describe vegetation properties. The findings of this study highlight the need to explore efficient ways to decrease
the complexity of multi-process-based calibration of land surface models arising from the increased dimensionality of both the
parameter and objective function spaces.
Finally, we note that more specific modelling objectives, such as the skillful representation of peaks flows (for flood forecasting
purposes), or low flows (for predicting summer drought impacts) could also be considered using the approach that has been
proposed. Similarly, the results and methods are applicable to other land surface models.
**Code availability**
Code of Efficient Elementary Effects (EEE) method is freely available with documentation and examples at
https://doi.org/10.5281/zenodo.3620895
**Competing interests**
The authors declare that they have no conflict of interest.



**Acknowledgments**

Financial support from the Canada First Research Excellence Fund and the Global Water Futures (GWF) program is gratefully acknowledged.

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
