# Peer review of "Towards reducing the high cost of parameter sensitivity analysis in hydrologic modelling: a regional parameter sensitivity analysis approach"

_Hydrology and Earth System Sciences, 2023_

## Author Comment (AC2)

**Towards reducing the high cost of parameter sensitivity analysis in hydrologic modelling: a regional parameter sensitivity analysis approach**

MS No.: hess-2023-21

**Response to reviewer comments**

The authors thank the reviewer for his comments and suggestions. In this document, we address the reviewer comments (RC) individually as follows.

**RC:** In this manuscript, Larabi et al. present a sensitivity analysis for 25 basins in North America using the VIC model. In addition to discharge, they used evapotranspiration and snow water equivalent as target variable for the sensitivity analysis. For an efficient simulation, they have clustered the basins based on catchment attributes. Based on the results of the sensitivity analysis, they distinguished into parameters that are overall informative / non-informative and into those that are relevant in a part of the catchments (variant-informative).
This study is well-written. All steps are clearly described. The results are supported by good figures.
Thus, I have only a few comments and can recommend it for publication.

**Response:** The authors thank the reviewer for his comments and suggestions. We addressed them as follows.

**RC:** L.38-50: I suggest to add that the parameter space could also be reduced by constraining parameter ranges to a smaller range.

**Response:** We thank the reviewer for his suggestion. However, we disagree that constraining parameter ranges to a smaller range would reduce the parameter space. It would affect the convergence rate of the search algorithm to the optimal solution but will not be helpful in reducing the number of calibration parameters which we address in the paper. We address this point in the discussion Line 519-521 as follows: '*Another approach to reduce the complexity of the calibration problem would be reducing the parameter ranges to a smaller range, which could speed the convergence rate of the search algorithm to the optimal solution. Hence it would reduce the computation time, but is bearing the risk of optimal values not being included in the too narrow ranges leading to false results (Mai, 2023).*'

Reference: Mai, J. (2023). Ten strategies towards successful calibration of environmental models. Journal of Hydrology, 620(A), 129414. http://doi.org/https://doi.org/10.1016/j.jhydrol.2023.129414

**RC:** 247: Sawicz

**Response:** done.

**RC:** 258: Line below the table is missing.

**Response:** We thank the reviewer for his comment. The line 258 is a space between the table and the new paragraph.

---

## Author Comment (AC3)

**Towards reducing the high cost of parameter sensitivity analysis in hydrologic modelling: a regional parameter sensitivity analysis approach**

MS No.: hess-2023-21

**Response to reviewer comments**

The authors thank the reviewer for his comments and suggestions. In this document, we address the reviewer's comment (RC) individually as follows.

**RC**: The manuscript 'Towards reducing the high cost of parameter sensitivity analysis in hydrologic modelling: a regional parameter sensitivity analysis approach' assesses the spatial pattern of parameter sensitivity and the performance of sensitive parameter transferability over 25 basins of the Pacific Northwest region of North America by using the VIC model. It is notable because a larger suite of 44 parameters parameter considered and the multiple output variables were assessed in the model. Overall, I think that the manuscript is well-written and the topic is attractive. My recommendation is that the paper be published in Hydrology and Earth System Sciences provided the following points are addressed by the authors.

**Response:** The authors thank the reviewer for his comments that are addressed as follows.

**RC**: I notice the authors clustered 25 basins and then test the ability of sensitive parameter transferring. To be honest, the number of basins used for clustering and parameter transferring (i.e., regionalization) is insufficient, you see, the basins of cluster #2 are only three in Figure 7. Insufficient samples will cause the classification results to show great randomness, thus impacting the parameter transferring.

**Response**: We thank the reviewer for raising this point. Although we have used a small number of basins, classification using the 25 basins yields a classification similar to that obtained when using a larger number of basins (i.e. the 158 basins that cover the entire large-domain) (see discussion Lines 492-499 and Figure 11). We have used a sample of 25 basins instead of the 158 basins to perform the sensitivity analysis due to computational limitations. The VIC model is computationally demanding with run time ranging between few minutes to few hours depending on the size of the basin. Therefore, doing the same analysis, which requires on average 430 model runs for each model output and each basin, on the 158 basins would be too time consuming.

**RC:** The result that two donor basins per cluster are sufficient to correctly identify sensitive parameters in a targeted basin, is doubtful. Have you compared the result to the previous research for parameter regionalization? As far as I know, at least using more than 5 donor basins is credible for parameter regionalization (e.g., Oudin et al., WRR, 2008; Bao et al., JH, 2012).

**Response:** Thanks for raising this question. We have not compared with previous research for parameter regionalization (e.g. Oudin et al., 2008; Bao et al., 2012) because we don't address the same issue. Previous parameter regionalization studies have mainly focused on transferring calibrated parameters to ungauged catchments, whereas here we focus on transferring parameter sensitivity information to other basins. The question concerns the choice of parameters to focus on in calibration. Clearly, the effort should focus on parameters that sensitively alter model behaviour, and thus the question is whether information that identifies these "sensitive" parameters can be transferred to other basins without performing further sensitivity studies for those basins. In addition, as shown in Oudin et al. (2008), the optimal of number of donor catchments is model dependent with performances decreasing when using more than five donor basins.

**RC:** Have you considered the cross-validation for the evaluation of the transferability of parameter sensitivity? I think cross-validation is a good way to check whether the sensitive parameter regionalization is reliable, i.e., using each of the basins in turn as if it were ungauged (Gou et al., BAMS, 2022).

**Response:** Indeed, we have used cross-validation to evaluate the transferability of parameter sensitivity (see Lines 267-286). Each basin of the 25 basins was set as a target basin. Then, informative parameters for each basin identified via the sensitivity analysis were compared to those identified by regionalization using the F1 score.

**RC:** This research does not involve parameter calibration, so what is the significance of sensitive parameter transferring? Why not transfer the calibrated parameters to ungauged basins directly?

**Response:** This paper does not address specifically parameter identification in ungauged basins. It rather addresses how to identify suitable candidate parameters for subsequent calibration over large domains at a reduced cost in general given the spatial variability of parameter sensitivity. Nevertheless, identifying parameter sensitivity over large regions could be useful in identifying suitable donor basins/regions and also in identifying which calibrated parameters could potentially be to transfer to ungauged basins.

**RC:** Line 93-Line 98: Check the data ranges again. 'average annual precipitation over the 25 basins ranges from 448 mm/year to 1666 mm/year.' But the average annual precipitation of basin 'BruneauR' is 337 mm. Meanwhile, you round to one decimal for temperature but keep two decimals for snow index and aridity index, need full text unified. Last, the definition of snow index should be merged with the caption of Table 2, i.e., here gives the calculation method and reference.

**Response:** Thanks for your comment. Appropriate modifications have been made in the text (Line 93-94). The definition of the snow index is provided with a reference as follows. '*The snow index, the fraction of annual precipitation that falls as snow when temperature is below 2°C (Woods, 2009; Sawicz et al., 2011 ),..*'.

**RC:** Line 172: 'Arno' to 'ARNO'

**Response:** done.

**RC:** Line 220: 'x' to '×'

**Response:** done.

**RC:** Line 220: What is the definition of elementary effects (EEs)? Is this value calculated based on the model output? Is the mean of the target modeled output?

**Response:** The elementary effect (EE) quantifies the change in model output $f(p)$ when a parameter $p_i$ is changed by a fraction of this parameter range $\Delta$. The elementary effect of parameter $p_i$ is calculated as follows:

$$EE_i = \frac{f(p_i + \Delta) - f(p_i)}{\Delta}$$

This definition is now included in the main text (lines 225-227).

**RC:** Line 239: 75 sets of noninformative/informative parameters are 75 sets of outputs or 75 experiments? Here 75 sets equal to the 25 basins multiply 3 model outputs. I think here cannot be represented as '75 sets of noninformative/informative parameters' because you adopted 44 parameters for each experiment, right?

**Response:** Thanks for your question. As mentioned in the manuscript (Lines 246-249),evaluating the sensitivity of the 44 parameters for each basin of the 25 basins and each output of the three model outputs, led to 75 EEE experiments. Each experiment required on average 430 model runs and converged to a set of noninformative/informative parameters. Hence, we obtain 75 sets of noninformative/informative parameters.

**RC:** Line 241: 'all 75 experiments' is right.

**Response:** Modified accordingly in text.

**RC:** Table1. The meaning of showing relief?

**Response:** Thanks for raising this question. As relief information was not used in this study, it has been removed from the table.

**RC:** Table2. The definition and method of the 'Snow Index' and 'Aridity index' should show in the methods section. There is no point in repeating the emphasis here. And the temperature threshold (i.e., 2°C) for snow index calculation, why? has any references?

**Response:** Thanks for your suggestion. The definition of the snow index and aridity index are removed from the table caption. The snow index definition and references are now provided in Lines 97-98 as follows: '*The snow index, the fraction of annual precipitation that falls as snow when temperature is below 2°C (Woods, 2009; Sawicz et al., 2011)*',...

**RC:** Table3. Again, the details of the table header, delete or move to the end of the table by 'Note'.

**Response:** Thanks for your suggestion. The details of the table are moved to the end of the table as a note.

**RC:** Figure 1. The basin ID can be marked.

**Response:** Thanks for your suggestion. The basin name is marked instead of basin ID in Figure 1 to help relate the basins when mentioned in the text with their location shown in the figure.

**RC:** Figure 2. How to distinguish whether the parameter is invariant-informative? 'Parameters are considered invariant-informative if the count of basins in which they are informative', how many informative basins are eligible for invariant-informative? 10 or 15?

**Response:** Parameters are considered invariant-informative if the count of basins in which they are informative equals 25 (i.e., informative at the 25 studied basins). The following definition was added to the Figure caption to define each parameter category. '*Parameters are considered invariant-informative if the count of basins in which they are informative equals 25, invariant-noninformative if that count is 0, and variant-informative if the count is between 1 and 24.*'

---

## Author Response (AR2)

**Towards reducing the high cost of parameter sensitivity analysis in hydrologic modelling: a regional parameter sensitivity analysis approach**

MS No.: hess-2023-21

**Response to reviewers comments**

The authors thank the Editor and the reviewers for their comments and suggestions. This document provides point-to-point replies to the reviewer 2 comment.

**Response to Reviewer 2**

**RC:** In addition to the first round, I have the following comments.

I am not convinced of the reply to my comment with regard to reduce the parameter space by constraining parameter ranges.

I agree with the authors that a reduction of the parameter range would not reduce the number of parameters. However, the title of the manuscript is related to a reduction of the high cost of parameter sensitivity analysis. Thus, overall the study is interested in more efficiency in parameter sensitivity analysis.

And reducing the parameter range (constraining parameters) would, as the authors mentioned in their reply and in the discussion, speed up the search algorithm. However, it reads in the discussion more as it is in principle an option, but is not recommended due to the risk that the optimal solution is not found due to a reduction of the parameter range.

However, a reduction of the parameter range certainly also reduces the parameter space. It is here more the question how to constrain parameter ranges in a reasonable way. Depending on expert knowledge of the model and information about catchment characteristics, a reduction of parameter ranges could reduce the computational cost of parameter sensitivity analysis.

Thus, with regard to more efficiency in parameter sensitivity analysis, it is required to mention all options. Therefore, I recommend the authors to improve this part of the manuscript and to include a more balanced statement.

**Response:** Thank you for your comment and suggestion. We mention the option of reducing the parameter ranges in the discussion section in a context of a process-based calibration where sensitivity analysis identifies a large number of parameters sensitive to the considered processes (see Lines 502-518). Hence, reducing the parameter ranges is discussed as an option to speed the

calibration process instead of improving the efficiency of the sensitivity analysis. We agree with the reviewer that a reduction of parameter ranges depending on expert knowledge of the model and information about catchment characteristics is useful but in a calibration context (i.e., parameter identifiability). In a sensitivity analysis, the computational cost is mostly tied to the number of parameters and the run time of the model (see Lines 42-44). For instance with the sensitivity analysis approach used here, 10N model runs were required on average with N being the number of parameters (see Line 213-214). Therefore, the computational time required is 10N x time required to run the model over a specific basin. Therefore, the parameter ranges would not make any difference to reduce the computational cost of the sensitivity analysis. This study is interested in efficiency of sensitivity analysis on large scales due to the computational cost of running hydrologic models over large domains particularly land surface models that have a large number of parameters. Specifically, the objective of the paper is to reduce the cost of identifying the spatial pattern of sensitive parameters over large domains at a reduced cost (see Lines 9-11). The Lines addressing reducing the parameter ranges are revised as follows (Lines 513-517):

*Another approach to reduce the complexity of the calibration problem would be to use a smaller parameter range, which could speed the convergence rate of the search algorithm to the optimal solution. However, this would have to be done carefully, possibly utilizing expert knowledge, in order to ensure the narrower range still contains the optimal solution (Mai, 2023).*